# Cardiorespiratory, Metabolic, and Performance Changes from the Effects of Creatine and Caffeine Supplementations in Glucose—Electrolyte-Based Sports Drinks: A Double-Blind, Placebo-Controlled Study

**DOI:** 10.3390/sports11010004

**Published:** 2022-12-22

**Authors:** Kunanya Masodsai, Thanachai Sahaschot, Rungchai Chaunchaiyakul

**Affiliations:** 1Faculty of Sports Science, Chulalongkorn University, Bangkok 10330, Thailand; 2Performance Plus Academy, Samutsakhon 74110, Thailand; 3College of Sports Science and Technology, Mahidol University, Nakonpathom 73170, Thailand

**Keywords:** sports drink, creatine, caffeine, isocaloric supplement, sprinting time, soccer performance

## Abstract

The purpose of this study is to investigate the additive effects of creatine and caffeine on changes in the cardiorespiratory system, metabolism, and performance of soccer players. Seventeen male soccer players randomly ingested three sports drinks comprising the following: glucose–electrolyte-based (Drink 1, control; D1), glucose–electrolyte-based drink + 5 g creatine (Drink 2; D2), and glucose–electrolyte-based drink + 5 g creatine + 35 mg caffeine (Drink 3; D3) during a 15 min recovery period after the modified Loughborough Intermittent Shuttle Test (LIST) on a standard outdoor soccer field. Then, a 20-m repeated intermittent sprinting activity was performed. The results showed no significant differences in cardiorespiratory and gas exchange variables. The non-significant levels of blood glucose concentrations among drinks with higher blood lactate concentrations were detected in parallel with increased heart rate during intermittent sprinting as a result of exercise intensities. Significantly longer sprinting time was found in D3 than D1 (*p* < 0.05), with no significant differences between D2 and D3. From this study, we conclude that the additive effect of caffeine–creatine supplements in a glucose–electrolyte drink during the 15 min recovery period enhances repeated 20-m high-intensity running in soccer players with no negative effect on cardiorespiratory functions.

## 1. Introduction

Since the creation of sports beverages, maintaining hydration status and performance has become a necessity, in addition to post-exercise recovery needs [1]. Sports drinks formulations are based on the extrapolation of the reductions in energy substrates and electrolyte loss from sweating. Therefore, sports drinks normally contain certain amounts of carbohydrates and electrolytes (sodium, potassium, calcium, and magnesium) [1]. Caffeine, the most popularly consumed psychoactive substance, is widely used not only by sedentary individuals but also among athletes due to known acute ergogenic effects on performance [2]. It has been shown that three out of four athletes consume caffeine before or during sports competitions [3]. Caffeine’s popularity was reported in a seven-year study that demonstrated 21% higher caffeine concentrations in urine from athletes consuming caffeine compared to the non-athlete control group [4]. Athletes from different sports use caffeine to improve strength and endurance, and caffeine drinks have been popularly consumed before or during sports competitions [3].

For active people, caffeine-containing products are available in different forms: drinks, chewing gum, gels, and bars [5]. In young active individuals, caffeine-containing energy drinks are quite popular [6]. Among athletes, the effects of caffeine-containing drinks on athletic performance exhibit a strong correlation not only with respect to endurance [2] and strength [7], but also with respect to cognitive functions [2]. Some reports raised awareness on the ergogenic effects of caffeine [8,9] in that several factors had to be considered, such as dosage, training intensity, ingestion time, time of day when consuming the caffeine supply, habitual caffeine consumption, and the proposed exercise type [8,10].

Creatine, a non-protein amino acid found in red meat [11], is another nutritional ergogenic aid that is popularly used among athletes to enhance strength and power [12,13]. The majority of creatines, known as the creatine pool (PCr + Cr), are stored in the skeletal muscle where their amount is limited to about 120 mmol/kg of dry muscle mass for a 70 kg individual. Intramuscular creatine is degraded into creatinine (a metabolic byproduct) and excreted in urine during a short and high intensity exercise bout [14]. Therefore, the body needs to replenish about 1–3 g of creatine per day to maintain normal (non-supplemented) creatine stores, which is particularly critical in some power-based athletes [15]. Without meat in regular meals, creatine supplementation becomes more serious in vegetarian athletes [16]. Due to the resynthesis of adenosine triphosphate (ATP) via the hydrolysis of PCr into Cr + Pi, this ergogenic substance helps maintain ATP availability, particularly during maximal effort anaerobic sprint-type exercises [17].

A normal regular diet contains only 1–2 g/day of creatine, which is not enough to maintain intense physical activity; therefore, the dietary supplementation of creatine is needed [14]. Creatine monohydrate is the most commonly used, where the uptake of creatine involves the absorption of creatine into the blood and then uptake is performed by the target tissues [18]; here, the peak of plasma creatine levels is sustained for about 60 min after ingestion [14]. Currently, creatine is popularly used by many athletes and military personnel because it is not banned by any sports organization [19,20,21].

Creatine supplementation increases intramuscular phosphocreatine concentrations, which may explain the observed improvements in high-intensity exercise performance, leading to higher acute anaerobic power [22] and greater training adaptations [23], especially in soccer players [24]. In elite female soccer players, the performance of various repeated sprint and agility tasks simulating soccer match play was acutely improved after consuming 5 g of creatine 4 times per day for 6 days [25]. The most effective way to increase muscle phosphocreatine storage is to ingest creatine monohydrates (0.3–0.8 g/kg /day) [26]. In one study, consuming a total creatine amount of 20 g/day for 5–7 days typically increased the total creatine content by 10–30% and phosphocreatine stores by 10–40% [23]. For safety reasons, it is recommended that individuals consume 3 g/day of creatine throughout their lifespan to promote general health [27].

Currently, sports drinks have been researched extensively but they still need to provide excellent alternative sources of energy for repeated high-intensity sports. The ergogenic properties of both creatine and caffeine among athletes have been separately identified. However, the acute additive effect is still in doubt. Based on sports drinks containing carbohydrates and electrolytes, it is crucial to examine changes in the cardiorespiratory system and metabolism in parallel to alterations in sports performance when creatine and caffeine are serially added.

## 2. Materials and Methods

### 2.1. Participants

Seventeen males (mean age 20 ± 2 years, height 1.75 ± 0.06 m, mass 70 ± 8 kg) were recruited from level 3 soccer clubs in Thailand’s professional soccer league. The inclusion criteria included age range between 18 and 25 years, regular participation in competitive soccer games, free from recent musculoskeletal injuries, no cardiorespiratory problems, and not taking medicine or supplements that could influence the cardiorespiratory system, metabolism, and exercise performance. Exclusion included those who had signs of any physiological complications or accidents during the experimental period. All players who fit the criteria completed a physical examination and a physical activity readiness questionnaire (PARQ) by a registered physical therapist and sports scientist. All participants were instructed to consume regular diets and to drink water ad libitum; refrain from vigorous physical activity; avoid all types of non-sports drinks, such as tea, coffee, and alcoholic beverages; and to sleep for at least 8 h for 2 days before the scheduled test date. This study was conducted during the off-season period and conformed to the Declaration of Helsinki and the regulations of the Institutional Review Board of Chulalongkorn University, Thailand (COA No. 650071). Prior to inclusion, comprehensive testing outlines and an explanation of the study were provided for participants. Written informed consent was received prior to conducting the physical tests.

### 2.2. Experimental Design

All participants completed three trials, separated by at least 5 days. Manufactured (Kovic International (Thailand) Co., Ltd. Bangkok, Thailand) isocaloric drinks that were electrolyte-based (placebo with 25 g glucose, drink 1, D1), electrolyte-based with creatine (drink 2, D2 with 25 g glucose and 5 g creatine monophosphate), and electrolyte-based with creatine plus caffeine (drink 3, D3 with 25 g glucose, 5 g creatine monophosphate and 35 mg caffeine) have a caffeine allowance (in the general drink) limited to no more than 15 mg per 100 mL per drink, according to the Thailand FDA (Ministry of Public Health, 2000). All three had identical taste, color, and flavor. Drinks were randomly provided during the 15 min recovery period in between the simulated soccer game. Participants were asked to record their food and fluid intakes, including the portion sizes of all foods consumed and the volume of all fluids consumed on the day prior to each trial in order to estimate their normal nutritional status.

### 2.3. Experimental Procedures

On test day, standing height, body mass, and body composition were measured using a free-standing adjustable stadiometer (IOI 353, Danilsmc, Gyeonggi-do, Korea) with a dry short and barefoot. Resting blood pressure was determined using an auto-sphygmomanometer (Omron, Kyoto, Japan). The hydration status was indirectly monitored using urine-specific gravity (USG; Refractometer DIP; model PEN-PRO; Atago, Tokyo, Japan). Using sterile techniques, blood samples from finger tips were collected for baseline glucose and lactate levels using an alcohol pad, sterilized single-use lancing devices (ACCU-CHEK safe-T-Pro Uno. Roche, Bern, Switzerland), glucose (ACCU-CHEK Active. Roche, Bern, Switzerland), and a lactate analyzer (Scout EKF Diagnostics, Cardiff, UK). Telemetry heart rates (Polar H10, Polar Electro Oy, Vantaa, Finland), the breath-by-breath gas analyzer (METAMAX 3B, CORTEX, Berlin, Germany), and a non-invasive hemodynamic monitor (Physioflow, Manatec Biomedical, Petit Ebersviller, France) were continuously monitored at rest, during, and post-exercise. Data at rest were recorded continuously while the subject sat quietly for 5 min and then completed a warm-up consisting of jogging, striding, and dynamic stretching for 10 min.

After this 15 min warm-up, the exercise protocol, a modified Loughborough Intermittent Shuttle Test (LIST), was conducted in a standard natural turf outdoor soccer field [28] from 8–11 AM. Briefly, participants completed three sets of 15 min of repeated running, walking, jogging, and sprinting between 20-m landmarks (first half; 3 × 5 min sets) separated by a 3 min seated recovery between sets. Body mass loss was obtained from differences between pre-first-half and post-first-half exercises. During the 15 min recovery, fluid replacements were separated into two portions: first randomly consuming 150% of body mass loss of either D1, D2, or D3, and then drinking water ad libitum. With the limited time, the latter ensured the highest possible amount of fluid replacements for soccer players. Thereafter, the second 45 min exercise session (second half) was conducted continuously with repeated intermittent sprinting and jogging between 20-m landmarks until fatigue. A subjective evaluation, which involved ratings of perceived exertion (RPE; 0–10 scale) [29], was also scaled throughout the period. Changes in performance were determined from sprinting time (ST) during the second half. All these tests were done on the same field and zone of the soccer club, where the natural turf was nicely looked after. This testing area was well managed to confirm the quality of the field, which may have affected the results. Moreover, the players were instructed to use the same shoes and sports wear. This study was conducted in a temperature range of 32.9–35.4 Celsius and 64–76% humidity.

### 2.4. Statistical Analysis

All data were presented as the mean and standard error of mean (SEM). When the homogeneity assumption of the variance was met (Leven’s test, *p* > 0.05), data were carefully analyzed using repeated-measure ANOVA with a Bonferroni post-hoc test by the SPSS statistical package (version 16; SPSS, Inc., New York, NY, USA). Statistical significances were accepted when *p* < 0.05.

## 3. Results

All participants were able to complete the first half of the test, but none of them could complete the 45 min sprinting exercise in the second half of the test. Anthropometric data and vital signs from three separate visits (Table 1) showed no significant differences among the resting conditions (*p* > 0.05). No significant differences regarding the amount of fluid replacement among the groups during the recovery period were found (*p* > 0.05).

Cardiorespiratory and metabolic functions (Table 2) show the exercise intensity-dependent characteristics in all groups. There was increasing cardiac, respiratory, and metabolic functions during the 3 × 5 min sets (S1, S2, and S3) during the first half, and these were even higher in the second half (fatigue) in all D1, D2, and D3 conditions. However, the data revealed no significant differences in these variables among the groups of either the first or the second half.

Blood glucose and lactate concentration changes showed similar patterns in all drinks without significant differences among the groups (Figure 1) at baseline, post-first half, and immediately after the second half exercises (*p* > 0.05). USG levels from resting to the end of the first half and at fatigue were 1.013 ± 0.002, 1.009 ± 0.003, and 1.024 ± 0.002 in D1; 1.008 ± 0.002, 1.009 ± 0.002, and 1.023 ± 0.002 in D2; and 1.015 ± 0.003, 1.013 ± 0.003, and 1.024 ± 0.002 in D3, respectively. Again, no significant differences in the USG levels either within or between groups were detected (*p* > 0.05).

Changes in aerobic capacities were tested in the second half from repeated high-speed running along the 20-m distance track (Figure 2). D3 shows a significantly longer ST compared with D1 (*p* < 0.05). However, there were no significant differences between D1-D2 and D2-D3 (*p* > 0.05). The RPE in all groups changed in similar progressively increasing patterns from baseline to fatigue among all trials.

## 4. Discussion

The present study shows enhancement of physical performance for sprinting time due to the combination of caffeine and creatine in glucose—electrolyte-based solutions. None of the drinks exhibited any changes in cardiorespiratory and metabolic responses during the first and second half of the tests.

Anthropometric data, resting blood pressures, and heart rate (Table 1) from the three tests were within normal ranges for young Thai males [30]. None of the participants showed abnormal signs or symptoms during the repeated tests. These tests were performed during the morning period during late summer season in Thailand, where the average ambient temperature and humidity were 32.9–35.4 Celsius and 64–76%.

All participants were able to complete the first 45 min test, but none of the participants could complete the second 45 min period test. The ranges were from 9.54 to 21.14 min in D1, 6.41 to 23.51 min in D2, and 11.31 to 37.51 min in D3. Similar intolerable results were shown as athletes performed physical activities in hot and humid environments [31]. RER during the second half were, in fact, nearly approaching 1.0. However, exercise termination was set up at the target HR of 90%of the age-predicted max HR.

Sports drinks represent palatable and efficient solutions for hydrating and supplying essential electrolytes, energy, and other nutrients employed and/or lost during physical exercise. The formulas can be designed according to the sports context: for example, to hydrate athletes, to restore energy, to improve circulation, and more recently, to improve mental focus and/or to prevent post-exercise muscle and joint pain [32,33]. The electrolyte-based drink is selected because electrolyte and water loss is the fundamental physiological response via sweating during physical activity [33]. Based on the electrolyte-based drink, the present study investigates physical performance changes when creatine and caffeine supplements are serially added.

Fluid and electrolytes replacements for those who will undertake the consecutive exercise session are compulsory. In professional soccer, fluid replacements are recommended both before and during the break [34]. Practically, volume replacements during recovery should exceed the volume that is lost during the first 45 min of exercise for two reasons: (a) to compensate for the ongoing water loss and (b) the additional sodium will assist fluid balance via extracellular osmolarity [35]. Assuming a 150% fluid ingestion during a short period of recovery, this amount of fluid may not be appropriate and not immediately exert any effect on blood volume and plasma osmolarity. We did not directly measure changes in blood volumes and osmolarities. However, the normal ranges of pre-exercise USG levels in the present study reflect the similar hydration status of all groups at rest. This will increase the hydration status at post-first half and post-fatigue.

In this study, the carbohydrate content of 25 g was added in all drinks. This follows the recommendation from a previous study that 22 g of carbohydrates every hour will improve physical performance [36]. To improve performance, ACSM recommends at least 30–60 g/h of carbohydrates during exercise [37]. Despite the amount of carbohydrates from previous studies, the present isocaloric supplemental designs for D1, D2, and D3 drinks will, therefore, similarly affect the physiological changes in this experiment. It is noted that more than 600 mg of caffeine will cause changes in cardiovascular functions [38], according to a previous study where maximum ventilatory responses were shown [39]. In the present study (Table 2), tidal volume, breathing frequency, and minute ventilation show the nearest maximum values. Thus, it is believed that the effect of high-intensity exercises on respiratory functions will overcome the effects of caffeine.

Steady blood glucose levels among healthy participants are expected from exercises with a similar duration and a similar level of intensity [40] in which high lactate concentrations reflect higher muscle metabolism using glycogen as energy substrates during endurance activities [41].

The exercise testing protocol in this study was modified from LIST. None of the soccer players in this study could run the full 45 min duration in the second half. The activity patterns of this specific test are proven to induce physiological and metabolic demands that are close to those observed in a soccer match [28]. A meta-analysis study concluded that exercise performance increased about 2% when 30–80 g/h of carbohydrates were added [41]. It is known that the exogenous carbohydrate will affect performance when muscle and liver glycogens became critically depleted from long-term physical exercise. A previous study indicated that muscle and liver glycogens are the main energy sources during moderate intensity exercise [42]. In addition, muscle and liver glycogen resynthesis takes about 6 hrs post-exercise, and is activated under the condition of 1 g/kg carbohydrate ingestion [43]. In the present study, a supplement of 25 g of carbohydrates was provided within the 15 min recovery session. This amount, as well as duration, may be inappropriate for completely energizing subsequent performance. This indicates that sprinting time improvements during the second half of the exercise are derived from other sources.

To provide adequate supplementations during recovery in order to improve the performance of the second half of the test in soccer players, creatine (D2) and creatine plus caffeine (D3) were serially added. The resynthesis of ATP via the hydrolysis of PCr helps maintain ATP’s availability, particularly during maximal effort anaerobic sprint-type exercises [17]. According to the very brief transit time in the gastrointestinal tract, almost 100% of creatine monophosphate is absorbed [44]. A study from muscle biopsy indicates that without creatine intake this process can replenish intramuscular ATP from 4–68% within 15 min under the condition of oxygen availability [45]. Thus, creatine supplementation alone may partly be involved with subsequent repeated high-intensity physical activity (Figure 2A). Over 70% of several hundred studies showed the remarkable positive effect of long-term creatine monohydrate, while the other 30% showed a small or insignificant effect [23]. The present study also indicated that acute creatine supplementation of 20 g/day for 5–7 days improves 5–15% of maximal power/strength, 5–15% of maximal effort muscle contractions (5–15%), 1–5% of single-effort sprint performance, and 5–15% of work performed during repetitive sprint performance. Receiving creatine monophosphate of 15 g/day for 5 days reduces the time needed to complete 40 m sprints [46]. In soccer players, creatine at 5 g/day ∗ 6 days improved 5- and 15-m sprint speeds, as well as jumping performance [47,48]. Doses, duration of creatine supplementation, and exercise tools cause the controversial results and need further investigations.

The effectiveness of caffeine on cognitive functions among athletes has been clearly demonstrated [49] in addition to endurance [2] and muscle strength [7]. The effects of caffeine on higher-center functions have been found [50] and considered an ergogenic substance on endurance performance [51]. Caffeine action is associated with adrenaline (epinephrine)-induced enhanced free-fatty acid oxygen-independent pathways and consequent glycogen sparing, which is closely related to anaerobic power [51]. The other mechanism explained that caffeine acts antagonistically on adenosine receptors, thereby inhibiting the negative effects of adenosine on neurotransmission, arousal, and pain perception. The hypoalgesia effects of caffeine resulted in dampened pain perception and blunted perceived exertion during exercise [52]. This will mitigate fatigue by extending the timepoint at which a level of pain would result in exercise termination.

A recent investigation on the parallel changes of electroencephalography (EEG) and cognitive functions shows the significant EEG alpha wave deactivation over the fronto-parieto-occipital areas following the consumption of a 50 mg caffeinated drink with an improvement of working memory observed in trail making A, B, and digit span tests [53]. A low dose of caffeine at 0.5 mg/kg body weight [54], which is equivalent to our study, should show the effect on cognitive function of soccer players.

The limitations of this study are the severe environment of 32.9–35.4 Celsius and 64–76% humidity and the repeated-use area of the field, all of which may affect players’ performance. The results obtained in the study were obtained at high humidity and temperature and it cannot be ruled out that with lower thermal stress, the results may be otherwise.

## 5. Conclusions

This study conducted on soccer players provides evidence that not only macronutrients but also psychoactive ingredients will potentially enhance performance, at least with respect to sprinting time. Practically, even hydration during recovery is the primary aim for sports drinks, and creatine (5 g) and caffeine (35 mg) will be another aspect of concern for athletes. They enhance performance with no harm on cardiorespiratory functions and can be safely used by athletes.

## Figures and Tables

**Figure 1 sports-11-00004-f001:**
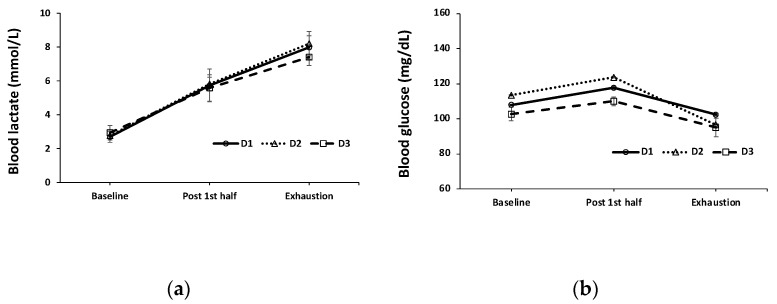
Blood lactate (**a**) and glucose concentrations (**b**) at baseline (resting prior to drink), immediately after the first half, and second half (fatigue) from D1, D2, and D3.

**Figure 2 sports-11-00004-f002:**
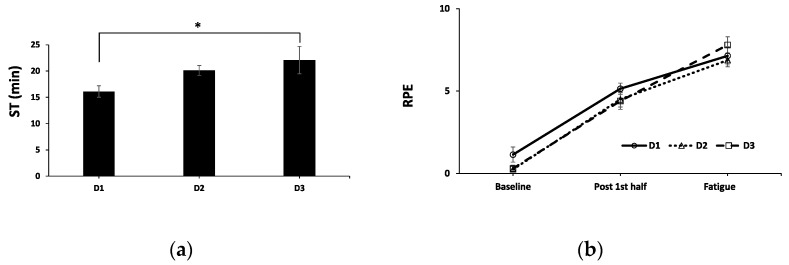
Sprinting time (ST, (**a**)) and the rate of perceived exertion (RPE; (**b**)) during the second half. While RPE increased in similar patterns, ST in D3 was significantly higher than D1. * *p* < 0.05, significant differences from D1.

**Table 1 sports-11-00004-t001:** Baseline anthropometric data and resting vital signs of participants (n = 17) during three visits prior to physical exercises and drinks.

Variables	Drink 1	Drink 2	Drink 3
Body mass (kg)	69.65 ± 2.18	70.06 ± 2.57	67.89 ± 2.38
Body mass index (km/m^2^)	22.82 ± 0.53	22.93 ± 0.60	22.14 ± 0.57
Body fat (%)	16.47 ± 1.12	15.35 ± 1.28	14.31 ± 1.26
Body muscle (%)	36.47 ± 0.34	36.41 ± 0.45	37.03 ± 0.46
Resting heart rate (bpm)	70 ± 2	72 ± 1	71 ± 2
Resting systolic blood pressure (mmHg)	119.21 ± 8.80	121.47 ± 8.87	124.47 ± 8.97
Resting diastolic blood pressure (mmHg)	71.49 ± 10.95	67.74 ± 11.65	70.87 ± 10.05

**Table 2 sports-11-00004-t002:** Cardiorespiratory and metabolic changes from soccer game simulations during the first half (S1, S2, and S3) and the second half (fatigue) with D1, D2, and D3.

Phases	Rest	S1	S2	S3	Fatigue
Drink 1					
Cardiac functions					
HR (bpm)	70 ± 2	139 ± 3	149 ± 3	155 ± 3	181 ± 2
SV (mL)	85.50 ± 1.99	117.98 ± 4.13	123.88 ± 5.10	126.13 ± 6.30	119.51 ± 5.54
CO (L/min)	6.25 ± 0.35	17.90 ± 1.06	19.97 ± 1.15	19.86 ± 1.49	21.90 ± 1.09
SVR (dyn·sec·cm^−5^)	1192.33 ± 87.23	437.66 ± 25.86	448.14 ± 59.82	409.27 ± 41.95	357.12 ± 13.65
Respiratory functions					
VT (L/time)	0.56 ± 0.04	1.70 ± 0.06	1.59 ± 0.05	1.58 ± 0.07	1.65 ± 0.05
BF (times/min)	18 ± 1	48 ± 2	54 ± 3	52 ± 3	60 ± 2
VE (L/min)	10.43 ± 0.82	81.78 ± 4.34	85.80 ± 4.05	80.91 ± 3.45	98.49 ± 3.45
Metabolic functions					
VO_2_ (L/min)	0.31 ± 0.02	2.79 ± 0.08	2.70 ± 0.08	2.59 ± 0.08	3.15 ± 0.08
VCO_2_ (L/min)	0.28 ± 0.02	2.70 ± 0.08	2.60 ± 0.07	2.49 ± 0.07	3.05 ± 0.08
RER	0.88 ± 0.02	0.97 ± 0.01	0.97 ± 0.02	0.96 ± 0.02	0.98 ± 0.01
EE (cal/h)	92.86 ± 8.82	830.69 ± 23.41	801.73 ± 23.25	770.42 ± 23.17	927.29 ± 24.74
Drink 2					
Cardiac functions					
HR (bpm)	72 ± 1	147 ± 3	159 ± 2	165 ± 2	186 ± 2
SV (mL)	89.94 ± 4.24	129.30 ± 9.44	123.92 ± 5.77	134.59 ± 11.98	118.94 ± 3.64
CO (L/min)	7.70 ± 0.49	20.51 ± 1.21	21.09 ± 0.99	23.22 ± 1.47	21.59 ± 0.76
SVR (dyn·sec·cm^−5^)	987.95 ± 75.88	402.95 ± 26.40	486.68 ± 79.60	399.60 ± 39.40	327.09 ± 12.66
Respiratory functions					
VT (L/time)	0.54 ± 0.04	1.74 ± 0.06	1.67 ± 0.07	1.63 ± 0.08	1.70 ± 0.07
BF (times/min)	19 ± 1	47 ± 3	49 ± 2	50 ± 3	58 ± 2
VE (L/min)	10.03 ± 0.80	79.78 ± 3.82	79.34 ± 2.78	79.84 ± 4.12	98.38 ± 3.62
Metabolic functions					
VO_2_ (L/min)	0.32 ± 0.01	2.96 ± 0.06	2.77 ± 0.06	2.59 ± 0.09	3.31 ± 0.07
VCO_2_ (L/min)	0.270.01	2.76 ± 0.09	2.62 ± 0.08	2.45 ± 0.09	3.11 ± 0.10
RER	0.85 ± 0.03	0.93 ± 0.02	0.94 ± 0.02	0.95 ± 0.01	0.97 ± 0.01
EE (cal/h)	91.58 ± 6.61	874.02 ± 20.07	819.80 ± 18.96	768.01 ± 28.00	957.23 ± 26.27
Drink 3					
Cardiac functions					
HR (bpm)	71 ± 2	145 ± 4	160 ± 4	162 ± 4	182 ± 3
SV (mL)	85.37 ± 3.62	122.89 ± 3.23	122.65 ± 3.83	132.02 ± 2.74	109.70 ± 2.92
CO (L/min)	6.42 ± 0.39	17.72 ± 0.64	20.17 ± 0.68	21.82 ± 0.83	19.93 ± 0.63
SVR (dyn·sec·cm^−5^)	1224.75 ± 90.01	446.83 ± 24.41	416.04 ± 30.72	425.59 ± 45.40	365.41 ± 13.06
Respiratory functions					
VT (L/time)	0.60 ± 0.04	1.82 ± 0.07	1.76 ± 0.05	1.65 ± 0.07	1.71 ± 0.06
BF (times/min)	19 ± 1	46 ± 2	48 ± 2	50 ± 3	56 ± 2
VE (L/min)	10.89 ± 0.74	81.33 ± 2.16	83.02 ± 2.66	80.80 ± 3.48	93.77 ± 2.03
Metabolic functions					
VO_2_ (L/min)	0.31 ± 0.01	2.92 ± 0.05	2.84 ± 0.05	2.58 ± 0.07	3.12 ± 0.06
VCO_2_ (L/min)	0.27 ± 0.10	2.75 ± 0.10	2.63 ± 0.10	2.51 ± 0.09	2.92 ± 0.12
RER	0.86 ± 0.02	0.94 ± 0.03	0.93 ± 0.03	0.97 ± 0.01	0.94 ± 0.03
EE (cal/h)	97.66 ± 5.95	865.49 ± 16.55	841.22 ± 16.24	768.91 ± 20.60	923.35 ± 19.70

HR, heart rate; SV, stroke volume; CO, cardiac output; SVR, systemic vascular resistance; VT, tidal volume; BF, breathing frequency; VE, minute ventilation; VO_2_, oxygen consumption; VCO_2_, carbon dioxide production; RER, respiratory exchange ratio; EE, energy expenditure.

## Data Availability

The data presented in this study are available from the corresponding author upon request.

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
