# Peer review of "Cardiorespiratory, Metabolic, and Performance Changes from the Effects of Creatine and Caffeine Supplementations in Glucose—Electrolyte-Based Sports Drinks: A Double-Blind, Placebo-Controlled Study"

_sports, 2022, doi:10.3390/sports11010004_

Round 1

Reviewer 1 Report

The title needs to be revised. It is creatine and caffeine content in a glucose-electrolyte drinks.

What is the rationale for adding creatine to a drink as for creatine to be effective in responders, there is normally a loading phase of a few days and maintenance phase after that. A rationale for creatine in a drink is not provided. Please provide justification in the introduction for the use of creatine.

Please provide the number of sprints in the three conditions following intake of the drinks.

In general, the abstract is not a good reflection of what was done and found in the study. See for specific comments below.

Please provide in the abstract the amount of creatine and caffeine present in the drinks.

L11. Aim in the abstract needs to be revised.

L15. Was it really a “simulated soccer game in the field” as it was intermittent sprinting. Please clarify.

L15. Please clarify “cardio-respiratory and gas exchange changes”. I suggest to mention the parameters for which there was no change.

L16. There was no effect on blood glucose so unclear what is meant by “Steady but insignificant levels of blood glucose concentrations”. Please clarify.

L21. I suggest to delete “and can be safely used for other endurance sports”.

L21. The term ‘harm’ seems to indicate that harm was expected. I suggest to revise.

L44. Ref 8 has no observations on caffeine. Please revise.

L84. Change “20 ± 1.67” to “20 ± 2” and “69.60 ± 8.17” to “70 ± 8”.

L91. Please clarify “or other activities”.

Ls 105-107. I suggest to present the data or at least clarify why this was done.

L113. USG is a measurement of hydration status. Please revise.

L125. Ref 29 is questionable.

L132. Please provide the recovery time between the sprints.

Table 1. Change “km.m2” to “kg/m2” or kg·m-2”.

Tables 1 and 2. Please express heart rate values without decimal places.

Table 2. Change “dyn/sec/cm-5” to “dynesseccm-5“.

Table 2. I suggest to express BF without decimal places.

Table 2. If the second half was maximal sprints. How confident are the authors on the metabolic observations as it would have expected that the RER would be above 1. Please discuss.

L172. I suggest to delete “Changes in”. Absolute values are plotted. I also suggest to alter the y-axis scale of glucose as values close to zero are unlikely.

L174. What were the instructions for the second half sprints and high-speed running is not the same as maximal speed running. Please clarify.

There are two Figures 2.

RPE was 6-20 according to L153 but it seems 0-10. Please revise.

L179. I suggest to delete “Changes in”.

L182. “some physical performance”. Please be specific.

L186. Please provide the blood pressure data.

L189. Please provided info environmental conditions. What was temperature and humidity.

Ls 200-202. Please provide some references.

L216. Please provide this information in the methods. Please also provide the composition of each drink.

L229. I suggest to consider glycogen as well as it seems that following intake of the drinks it was mainly high-speed running.

L233. TTE, but for what exercise modality. Please revise.

L237. “can be almost completely exhausted within 45-90 min”. Time is not correct and muscle glycogen was lower but definitely not exhausted. Please revise.

L249 “However, this process can replenish intramuscular ATP for 4-68% within 15 min under the condition of oxygen availability [47]”. As far as I know, this is not the result of creatine intake. Please revise. In addition, ref 47 was on isolated muscle tissue so the in-vivo generalisation may be not that clear.

L254. The authors need to provide observations from the literature that there can a be an acute effects of the amount of creatine used in the present study.

L254. “This study”. Please clarify which study.

Ls 252-259. To use observations from repeated intake of creatine to just single use in the present study is problematic. This section needs substantial revision.

L279. Not clear what is meant by “mental-activated ingredients”.

References. Please consult author guidelines as journal names are provided with full names and abbrevations.

References. Please change “doi:https://doi”

Reviewer 2 Report

Dear Authors:

Interesting article about intervention, However, some changes are proposed

Line 7: Delete line behind email

Line 17: "higher lactate concentrations", May be blood lactate..?

Line 26-27: "Since sports beverages had been started, there has been the necessity of maintaining hydration status and performance, and later for post-exercise recovery needs". This sentence needs a reference

Line 84: "weight", better body mass

Line 84: 17 players, 3 groups..?. Did you ckeked the G- power, IN ORDER TO ACCEPT THE STATISTICAL POWER

Line 97: Include Helsinki Declaration

Line 100: Manufactured isocaloric drinks. Include Company name

Line 105: "Subjects", better participants

Line 109: Please inlcude Resgister in all products

              Did you concreted warm-up?

Line 110: "weight", better body mass

Line 128: "weight", better body mass

Line 130: "weight", better body mass

Line 134: Subjective evaluation, ratings of perceived 134 exertion (RPE; 6 -20 Borg's scale). Include the reference.

Line 138: Did you cheked the Levene test, before ANOVA? and before T-test did you ckecked the nomality of the data? And interaction among groups? Why did you correlated some dependent variables with RPE?, feel in sport elite is very common concept.

Tables. Include P-value 

           T-test, Did you inclide effect size..?

Line 144,189,190: "Subjects", better participants

Line 151: "Weight", better body mass

Line 162: 2 in sub-index

Line 181: Discussion: Incluce the main goal

Line 186: Heart rate (HR)

Line 194-5: Sports drinks represent the palatable and efficient solutions to hydrate and supply essential electrolytes and energy, and other nutrients employed and/or lost during physical exercise

Line 201: This sentence needs reference 

nácio SG, de Oliveira GV, Alvares TS. Caffeine and Creatine Content of Dietary Supplements Consumed by Brazilian Soccer Players. Int J Sport Nutr Exerc Metab. 2016 Aug;26(4):323-9. doi: 10.1123/ijsnem.2015-0134. Epub 2015 Dec 16.

Line 208: "Considering on 150% fluid ingestion, even was recommended,  during a short period, 15 min, of recovery in soccer, this amount of fluid may not be appropriate and not immediately exert any effect on blood volume and plasma osmolarity. In our study, we did not measure changes in blood volumes and osmolarities. However, the normal ranges of pre-exercise USG levels minimize confounding factors and maxim  izes measurement reliability of later USG values. In addition, no significant differences of USG levels for both within and between groups, indirectly, reflect the similar hydration status of all groups at rest, post-first half, and post-exhaustion". I DON THINK THAT THIS PARAGRAPH IS CORRECT, REWRITE AND NEEDS MORE STRONGTH ARGUMENT.

Line 274: "With similar 274 dosage (35 mg) in this study, the duration of 15 min recovery should show the effect of caffeine on cognitive function of soccer players.".... Needs reference 

References: Include this recent review.

Mielgo-Ayuso J, Calleja-Gonzalez J, Marqués-Jiménez D, Caballero-García A, Córdova A, Fernández-Lázaro D. Effects of Creatine Supplementation on Athletic Performance in Soccer Players: A Systematic Review and Meta-AnalysisNutrients. 2019 Mar 31;11(4):757. doi: 10.3390/nu11040757.

Line 369: TESTING FOR MULTIPLE BUBBLES: HISTORICAL EPISODES OF EXUBERANCE AND COLLAPSE IN THE S&P 5TESTING. CAPITTAL LETTER?

In Advanced

King Regards

Round 2

Reviewer 1 Report

In the abstract, please clarify that the LIST (at least half of a LIST) was performed before the recovery.

In the abstract, please clarify the distance of the sprints.

In the abstract, state that there were no differences in blood glucose. Mention of insignificant is akward.

In the abstract, the lactate observations are still unclear in which condition they were higher. Please clarify.

L22. Change “with insignificant differences” to “with no differences”.

In the abstract, the conclusion needs to mention that the creatine and caffeine were included in a glucose-electrolyte drink.

I suggest to clarify “enhances the physical performance” by “repeated 20 m high-intensity running. A sprint is normally performed with all-out maximal efforts but that does not seem to be case based on the revisions you made to the first draft.

In the abstract, please reconsider “metabolic functions” as only data on glucose and lactate are provided.

L74. I suggest to change “increases intramuscular creatine concentrations” to “increases intramuscular phosphocreatine concentrations”. See also L77. The increase in intramuscular phosphocreatine is associated with enhanced exercise performance.

In the introduction, there is no mention of a potential acute effect by creatine. When that information is not available then please state that.

L94. Change “69.60 ± 8” to “70 ± 8”.

Ls 113-116. Please provided the glucose content of the drinks.

L151. According to your response, task failure was not at exhaustion by at 90% of age-predicted maximal heart rate. Please revise. Also, Table 2 has “Exhaust” which is not correct. Please change. If the test was stopped at 90% of age-predicted, why is it that heart rate in Table 2 at “exhaust” for the 3 conditions are not similar. Please clarify and revise.

Table 1. 1.83 needed to change to 2. In addition, in Table 2, rounding to the nearest absolute number has been done incorrectly. Please revise.

L 213. Please provide the blood pressure data in Table 1.

Ls 211-212. Were drinks consumed before the 15-min recovery? Please clarify

L271. Please provide composition information in the methods.

L288. It was not volitional exhaustion. This needs to be clear throughout the manuscript. TTE when referring to the second half exercise and is not correct either. Check throughout the manuscript that TTE is only used when appropriate.

L292. TTE, but for what exercise modality. Was this for repeated high-intensity running. Please clarify whether the exercise modality was similar as in the present study.

Reviewer 2 Report

Accepted

Author Response

Thank you so much.